# Half a Century of Fragmented Research on Deviations from Advised Therapies: Is This a Good Time to Call for Multidisciplinary Medication Adherence Research Centres of Excellence?

**DOI:** 10.3390/pharmaceutics15030933

**Published:** 2023-03-14

**Authors:** Przemysław Kardas, Tamás Ágh, Alexandra Dima, Catherine Goetzinger, Ines Potočnjak, Björn Wettermark, Job F. M. van Boven

**Affiliations:** 1Medication Adherence Research Center, Department of Family Medicine, Medical University of Lodz, 90-419 Lodz, Poland; 2Syreon Research Institute, 1145 Budapest, Hungary; 3Center for Health Technology Assessment and Pharmacoeconomic Research, University of Pécs, 7623 Pécs, Hungary; 4Fundació Sant Joan de Déu, 08007 Barcelona, Spain; 5Deep Digital Phenotyping Research Unit, Department of Precision Health, Luxembourg Institute of Health, Strassen, 1445 Luxembourg, Luxembourg; 6Faculty of Science, Technology and Medicine, University of Luxembourg, Esch-sur-Alzette, 4365 Luxembourg, Luxembourg; 7Institute for Clinical Medical Research and Education, University Hospital Center Sisters of Charity, 10000 Zagreb, Croatia; 8Department of Pharmacy, Faculty of Pharmacy, Uppsala University, Husargatan 3, 752 37 Uppsala, Sweden; 9Faculty of Medicine, Vilnius University, Universiteto g. 3, LT-01513 Vilnius, Lithuania; 10Department of Clinical Pharmacy and Pharmacology, Medication Adherence Expertise Center of the Northern Netherlands (MAECON), University Medical Center Groningen, University of Groningen, 9713 GZ Groningen, The Netherlands

**Keywords:** medication adherence, centre of excellence, research quality, advocacy, medical education, evidence-based medicine

## Abstract

Medication adherence is a key precondition of the effectiveness of evidence-based therapies. However, in real-life settings, non-adherence to medication is still very common. This leads to profound health and economic consequences at both individual and public health levels. The problem of non-adherence has been extensively studied in the last 50 years. Unfortunately, with more than 130,000 scientific papers published on that subject so far, we are still far from finding an ultimate solution. This is, at least partly, due to fragmented and poor-quality research that has been conducted in this field sometimes. To overcome this deadlock, there is a need to stimulate the adoption of best practices in medication adherence-related research in a systematic way. Therefore, herein we propose the establishment of dedicated medication adherence research Centres of Excellence (CoEs). These Centres could not only conduct research but could also create a profound societal impact, directly serving the needs of patients, healthcare providers, systems and economies. Additionally, they could play a role as local advocates for good practices and education. In this paper, we propose some practical steps that might be taken in order to establish such CoEs. We describe two success stories, i.e., Dutch and Polish Medication Adherence Research CoEs. The COST Action “European Network to Advance Best practices & technoLogy on medication adherencE” (ENABLE) aims to develop a detailed definition of the Medication Adherence Research CoE in the form of a list of minimal requirements regarding their objectives, structure and activities. We hope that it will help to create a critical mass and catalyse the setup of regional and national Medication Adherence Research CoEs in the near future. This, in turn, may not only increase the quality of the research but also raise the awareness of non-adherence and promote the adoption of the best medication adherence-enhancing interventions.

## 1. Introduction

With an increasing number of medications being developed and available, medication adherence has become more important than ever. Indeed, an adequate level of adherence to evidence-based pharmacotherapies is a sine qua non precondition of their high effectiveness in real-life conditions, similar to those observed in optimal conditions of highly controlled clinical trials (i.e., efficacy). 

Unfortunately, non-adherence to medication is still very common. According to a seminal World Health Organisation report published in 2003, around 50% of patients deviate from their prescribed chronic treatments [1]. Little improvement has been observed in the 20 years since [2]. Therefore, this unfortunate scenario still leads to poor health outcomes, increased morbidity and mortality at the individual level, as well as increased use of health services and higher costs at the health system level [3,4].

On top of all these, there are new reasons to look at medication adherence with special care. Intensified ageing of the global population leads to frequent multimorbidity and related polypharmacy, which is often the cause of non-adherence [5]. Another new barrier to proper adherence to chronic treatments has been built by the recent outbreak of the Covid-19 pandemic. This unexpected disaster formed a sort of stress test for healthcare systems and proved that under such circumstances, many of them were not able to secure full maintenance of chronic treatments for their beneficiaries, which made medication adherence a real challenge [6,7]. 

At the same time, the literature observes an increasing interest in adherence-related research. The total number of publications, including the terms ‘medication adherence’ or ‘patient compliance’ (the equivalent term that has been used more broadly in the past), being available in the Medline (via PubMed) database, currently exceeds 130,000, and it is still growing (Figure 1). This, however, is not fully reflected in clinical practice, where more than 50 years of dedicated research has not yet led to any breakthrough. In consequence, medication adherence rates remain suboptimal, and many healthcare professionals do not want to admit that medication non-adherence is a problem nor to implement available interventions [8].

Major obstacles hindering progress in this area include fragmentation of adherence research, often using mono-disciplinary, non-standardised approaches, which results in a scarcity of high-quality output. In the last decade, several solutions for increasing standardisation and quality have been proposed. In 2012, the consensus ABC terminology and taxonomy of medication adherence were introduced [9], followed by the publication of the EMERGE guidelines on reporting adherence-related studies in 2018 [10]. However, neither key indicators for assessing the effectiveness of adherence-enhancing interventions nor standard procedures of their benchmarking are agreed upon. They would allow for objective selection and scaling up of the most effective and cost-effective interventions. This is particularly true for the digital area, such as eHealth technologies which have been developing rapidly. However, their full potential is still not fully understood nor utilised [11]. Notably, non-adherence remains overlooked in national research and policy agendas, as pointed out by the dedicated OECD report [12]. Last but not least, poor-quality research leads to ineffective spending on both studies and intervention delivery. In particular, reimbursement decisions related to these interventions require high-quality evidence of their effectiveness and cost-effectiveness, which is currently lacking quite often. In consequence, the number of adherence-enhancing interventions reimbursed in Europe is still very low [13,14].

In such circumstances, it is necessary to join efforts focused on adherence research and stimulate the adoption of best practices in medication adherence-related research in a systematic way. In practical terms, it means that there is a need for the establishment of national or regional Medication Adherence Research Centres of Excellence (CoE). These Centres could not only conduct research but could also create a real societal impact, directly serving the needs of patients, healthcare providers and systems, as well as economies. Additionally, they have the potential to merge stakeholders from different backgrounds together in order to set national priorities and strategies addressing the issue of medication non-adherence. Thus, they could play a role as local trendsetters and advocates for good practices and education. 

Stimulating the creation of Medication Adherence Research CoE is one of the key objectives of a novel European scientific collaboration, launched in 2020 under the name of “European Network to Advance Best practices & technoLogy on medication adherencE” (ENABLE). ENABLE is a COST Action funded by the European Commission, which brings together researchers from 40 European countries and Israel. Other principal objectives of ENABLE include raising the awareness of adherence-enhancing solutions, fostering and extending multidisciplinary knowledge on medication adherence at patient, provider and system levels, accelerating the transfer of this knowledge from producers to useful clinical application and developing collaboration towards an economically viable policy and implementation of adherence-enhancing technology across different European healthcare systems [2]. Analysing all these aspects, we will highlight the potential benefits of setting up CoEs for medication adherence, illustrated by examples and best practices from across Europe.

## 2. Scope of the Centre

In general, CoEs are organisations that aim for the highest attainable standards in their specific fields by synergies created through an exceptionally high concentration of expertise and related resources centred on a particular area. By meeting specific guidelines related to outcomes, process, volume and infrastructure, as well as the application of innovative tools, technologies and techniques, they can deliver enhanced quality of their output [15].

Various types of CoEs exist in the field of medicine. Based on the main area of their activity, the following major types may be distinguished:Educational centres;Clinical or health care centres;Research centres, which usually provide some educational and/or clinical services; however, research remains their dominant activity [16].

Hereby, we call for the establishment of medication adherence CoEs that belong to the last of the above-mentioned categories. Such medication adherence research CoEs should play multiple roles (Table 1). Those of primary importance are related to research, such as performing high-quality studies on adherence, disseminating results and promoting good practices in adherence science. In this regard, it is essential for medication adherence CoEs to collaborate with various funding agencies, patient organisations, governments and other stakeholders and encourage the use of available resources for research conducted in compliance with the best available standards [17], such as the above-mentioned ABC taxonomy and EMERGE reporting guidelines.

Along with these principal aims, other areas of CoEs’ activities are worth considering, such as clinical practice, e.g., promoting the use of the best evidence-based interventions and implementation strategies and introducing novel technologies for medication adherence management. Due to the accumulation of unique knowledge and experience, CoEs will be able to provide professional education to healthcare providers in medication adherence management, responding to the well-defined need of this group of stakeholders [8]. Similarly, CoEs are perfectly placed to provide professional communication to the general public and, last but not least, advocacy. Many of these aims are interdisciplinary, such as the development and adaptation of high-quality, evidence-based clinical practice guidelines, as well as the tools for the dissemination and implementation of this guidance, with the ultimate aim of translation of medication adherence interventions into clinical practice [17,18]. Therefore, in the case of medication adherence research, a CoE’s interdisciplinarity is not simply a challenge but a fundamental necessity.

Due to its unique competencies, a medication adherence research CoE should also work on strategic thinking and forecasting. Across all areas of its operation, it should stimulate creativity and innovation, thus establishing direct industry and economic connections [16].

## 3. How to Create a Centre of Excellence?

Below, we provide some general suggestions on how to create a medication adherence research centre of excellence. Current guidance on how to create a CoE is somewhat limited. Therefore, one part of this recommendation is the result of theoretical analysis, whereas the other is based on real-life experience and describes two European success stories.

Due to the scarcity of scientific literature on research CoEs, it seems reasonable to learn from the experience of other types of CoEs and similar institutions [17,19]. We found it useful to adopt a framework which originally had been designed for the development of a clinical CoE [15]. This framework is based on a 3-stage protocol, as illustrated in Figure 2: 

### 3.1. Stage 1: Vision and Validation

Organisation committee members must first assess the readiness of the environment and/or an umbrella institution (e.g., a university) to operate the given centre by verifying the availability of human, technical and financial resources, the operating business model, organisational culture and leadership support. If all these conditions are met, efforts of the committee are then directed towards designing a working mission and vision statement. A suggested vision, mission and values for a medication adherence research CoE are presented in Box 1.

Box 1Suggested mission, vision and values of medication adherence research centres of excellence (adapted from [17]).
MissionTo design, execute and disseminate high-quality research in the area of medication adherence.VisionMedication Adherence Research CoE will be the leading regional/national source of high-quality evidence-based recommendations to guide scientific research, clinical practice, healthcare system organisation, and education in the area of medication adherence.ValuesGood research practice;Innovation.


### 3.2. Stage 2: Design and Development

As soon as the stage of conceptualisation is completed and feasibility is verified, detailed operational plans need to be prepared. They should address the components of organisation design, area of future activity, staff, venue, relationship with an umbrella organisation, visibility, finance, etc. 

### 3.3. Stage 3: Completion and Start of Operation

After the plans are executed, and formal authorisation is obtained (if required), a centre of excellence is launched and starts to operate according to adopted protocols.

Along this process, and particularly at Stages 1 and 2, there arise a number of important questions that need to be answered, such as:Affiliation—although a non-academic affiliation is potentially possible, it seems that a direct link to academia is a natural scenario for a research-oriented CoE. This, however, may create another challenge, as universities usually do not have established policies for developing centres of research excellence. Moreover, they may insist on refocusing the major area of CoE activity from research to education.Infrastructure—is it possible for CoE to obtain its own venue, equipment etc., or will it rather be a ‘virtual’ institution or platform within the umbrella organisation that shares or reuses its infrastructure?Budget—is it possible to guarantee a certain budget for the centre, or is it rather going to be funded by its activities, e.g., research grants, expert opinions, training, etc.?

## 4. Success Stories

Despite high interest in medication adherence and a great amount of medication adherence-related research (see Figure 1), only a few dedicated research centres exist in Europe. Perhaps, one of the reasons is an interdisciplinary dimension of medication adherence, which does not ‘belong’ to one single area, e.g., cardiology, respiratory medicine, epidemiology, pharmacoeconomics, nor even to one single discipline, e.g., medicine or pharmacy, etc. Nevertheless, existing centres are worth a closer look at as they are a potential source of inspiration for setting up other centres.

## 5. Dutch Experience

The Medication Adherence Expertise Centre of the Northern Netherlands (MAECON) was established in 2018 and had its seat at the University Medical Centre Groningen/University of Groningen in the Netherlands. Seeing the diverse challenges and being unable to solve the issue of medication non-adherence on their own, the original founders recognised the need for multidisciplinary collaboration on the topic of medication adherence. Therefore, this initial faculty of MAECON involved Principal Investigators (PIs) representing various disciplines, including different medical specialities: general practice, pharmacy, bioanalysis, technology, epidemiology, sociology, psychology and education. Its steering committee is composed of the heads of the clinical departments dealing with chronic diseases at the hospital, and its assets include clinical trial support within the hospital, a state-of-the-art bioanalytical lab (for assessment of drug concentrations in blood, urine, sputum and scalp hair), rich clinical and pharmacy prescription databases and a network in primary care and community pharmacy in the outreach area of the hospital. Its four pillars include research, education, clinical practice and private-public partnerships. MAECON celebrated its launch with a symposium on medication adherence and the establishment of a national medication adherence day in the Netherlands (18 November). Several media, including newspapers, radio and television channels, had news items on the launch of MAECON. Over the years that followed, MAECON has implemented several research projects related to medication adherence, including clinical trials with innovative digital technologies, such as digital inhalers, electronic spacers [20], smart pill bottles and blisters [21], as well as novel bioanalytical essays for measuring adherence in body fluids [22] and practical adherence-enhancing tools have been studied and developed [23]. Additionally, several systematic reviews on effective interventions were performed [24]. MAECON PIs has led nationally and commercially funded adherence grants and a national integrated care pathway for adherence to respiratory medications. Furthermore, different educational courses and webinars were held for students and healthcare professionals. Social media channels, websites, articles in patient magazines and editorials were used to raise awareness of medication adherence [25,26,27]. Last but not least, MAECON is currently leading the European Commission-funded COST Action ENABLE, focused on the economically viable implementation of technologies enhancing medication adherence across different European healthcare systems [2].

## 6. Polish Experience

The Medication Adherence Research Centre (MARC), affiliated with the Department of Family Medicine of the Medical University of Lodz, Poland, was created under the Order on 23 December 2020 by the Rector of the University. This step was a conclusion of over 20-year-long commitment of the local team to research in the area of medication adherence. The milestones of that process were the first randomised controlled trials in that area conducted in Poland [28,29,30,31,32], the first Polish doctoral thesis focusing on this topic [33], and finally, the coordination of the ABC Project, a unique European research collaboration devoted to medication adherence, conducted under the 7th Framework Program [34]. Among many effects brought by that project, one particularly worth mentioning is a consensus terminology and taxonomy of medication adherence, which now constitutes a worldwide-recognised basis for research in that area [35]. 

The interdisciplinary nature of medication adherence deserves a multifaceted approach. Therefore, within the short time from the formal creation of MARC, several aspects of this issue have been addressed in its activities. Accepting that the demographic changes occurring in both Poland and all of Europe led to interlinked problems of polypharmacy and non-adherence, comprehensive guidance on polypharmacy management in elderly patients has been prepared [5]. Furthermore, within the framework of the dedicated Skills4Adherence project, coordinated by the MARC’s team, an online educational program on polypharmacy and adherence management in the elderly has been designed and made freely available in four languages, including English [19]. Currently, it is extensively used by European healthcare students and professionals [36].

Acknowledging the active role of the patients in the therapeutic alliance, MARC devoted its activities to the study assessing the patient perspective, particularly when it comes to patient-centric pharmaceutical product design [37]. A discrete choice experiment conducted by the centre’s team allowed for the assessment of Polish patients’ preferences and willingness to pay for various forms of oral medications [38].

Finally, due to the rising role of digital health, MARC advocated the use of available Big Data for the assessment of medication adherence at the global forum [11]. At the national level, its team actively lobbied for the introduction and the wide use of ePrescriptions in Poland, anticipating benefits from this scenario for adherence research. Indeed, as soon as ePrescriptions became available, the first nationwide assessment of primary non-adherence was made [39], and a comprehensive nationwide analysis of polypharmacy was conducted [40].

The formal launch of MARC came along with the start of the ENABLE network. The Polish team took an active part in this cooperation, organising the first physical ENABLE Working Group meeting in Lodz, Poland, on 16–17 September 2021. Moreover, MARC initiated two important studies within the framework of ENABLE, i.e., TherapyMaintenance@Covid, which assessed the medication management practices in place for chronic diseases during the COVID-19 pandemic across European countries [6,7], and the EUREcA study which identified reimbursed medication adherence enhancing interventions currently available in Europe [13,14]. 

## 7. Lessons Learned

The above-described CoEs are subject to challenges that the pioneers faced. A structured discussion that took place recently on the forum of ENABLE group highlighted some of them that may be of particular interest to future CoEs. These challenges are listed below:(1)Is it necessary to meet specific predefined criteria for the creation of CoE? Both above-described CoEs benefit from the long-term commitment of their founding members to the area of medication adherence research, and they have been shaped according to an ‘organic’ evolutionary process. Therefore, their design is rather the consequence of locally available resources and historical reasons and is not necessarily informed by a set of predefined criteria. Of note, for medication adherence research CoEs, such criteria are not agreed upon yet. However, setting these criteria might be useful for potential new CoEs to come. Outside the medication adherence area, there are a few good practices that can serve as inspiration. For example, in software engineering, an assessment approach has been developed, which uses established scoring criteria and standards to certify an organisation [41].(2)How to secure a sufficient budget for the CoE? Adopted solutions included making as much as possible use of facilities existing in hospitals/academia, searching for external grants, and attracting both public and private funding.(3)How to assure good cohesion among CoE members if the CoE works across departments, within and outside the hospital? The way of addressing this challenge was to set up an efficient management structure, assure transparent communication, and complement each other’s expertise without rivalry.(4)Last but not least, how to increase the visibility of the CoE across the country and beyond? Actions taken to address this aim included press releases, organising a National Day of Medication adherence, creating an attractive website, scientific publications in high-impacted journals (using the affiliation of the CoE), and other methods of active targeting of different stakeholders, such as pharmaceutical and MedTech industry, healthcare professionals, patients, etc.

## 8. Looking Ahead

A critical analysis of the strengths and weaknesses of the two European medication adherence research CoEs described above allows the lessons learned to be used in the future. So far, there have been no overarching provisions developed that would specify what may or may not be qualified as a medication adherence research CoE. There exists no standard definition for it either. Under such circumstances, there is a risk that institutions that are not truly excellent may be labelled as such, diluting the meaning of CoE and creating confusion [42]. Thus, an alternative method involves the selection of a central organisation to develop standardised criteria and approve the quality of CoE [43].

Due to the limited lifetime of the ENABLE COST Action, which is planned for four years (2020–2024), it is not possible for this collaboration to become an approving and licencing body, which would certify new emerging centres as medication adherence research CoEs. However, designing guidelines, publishing them in the form of a roadmap and distributing it among stakeholders is a good way to promote the establishment of CoEs [16]. Moreover, encouraging and supporting the creation of such centres lies within the scope of ENABLE’s core objectives [2]. Therefore, ENABLE initiated an internal discussion on CoEs, which led to the creation of this very publication. Moreover, future work within ENABLE will be focused on the development of more detailed definitions of the medication adherence research CoE. It will take the form of a list with minimal requirements regarding research, as well as other areas, such as clinical practice, education, advocacy and policy, etc. When formulating these requirements, a balance must be kept between the necessity of promoting high research standards on the one hand and, on the other hand, the importance of promoting an inclusive approach to medication adherence research in countries where the interest in this topic is still in its early stages.

Currently, successful attempts have been made to link organisations with a similar scope of activity, such as the International Society for Medication Adherence (ESPACOMP) and the International Society for Pharmacoeconomics and Outcomes Research (ISPOR), and particularly, ISPOR’s Medication Adherence and Persistence Special Interest Group. Such a concept gives hope for developing generally accepted standards.

We do expect that this publication will help to create a critical mass and catalyse the setup of regional and national medication adherence research CoEs in the near future. This, in turn, may be followed by the creation of an international network of CoEs, which, undoubtedly, is a very ambitious, yet not impossible, scenario. Many successful initiatives have been carried out so far by a unique platform that has been created by ENABLE. Additionally, discussions on the issue have already been opened in several countries (e.g., Switzerland, France, Spain, and Hungary), which gives a good prospect for the planned undertaking.

## Figures and Tables

**Figure 1 pharmaceutics-15-00933-f001:**
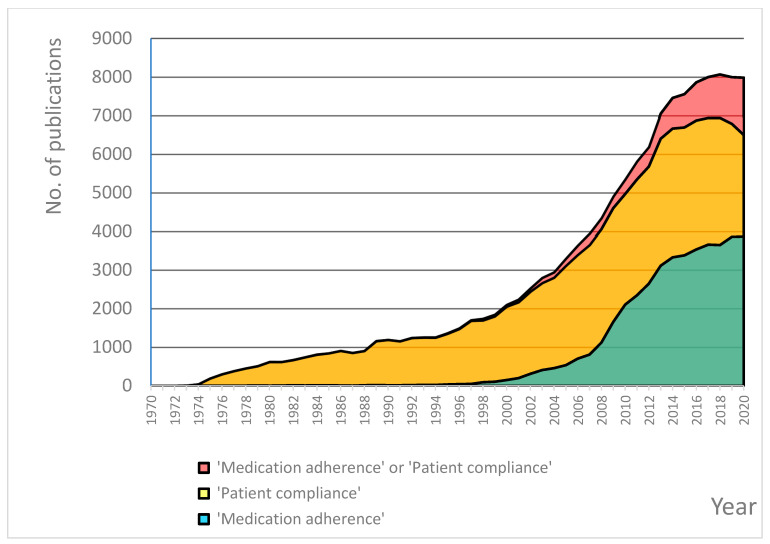
The yearly numbers of scientific publications on medication adherence, by keyword used, retrieved by the PubMed online database for the period 1970–2020.

**Figure 2 pharmaceutics-15-00933-f002:**
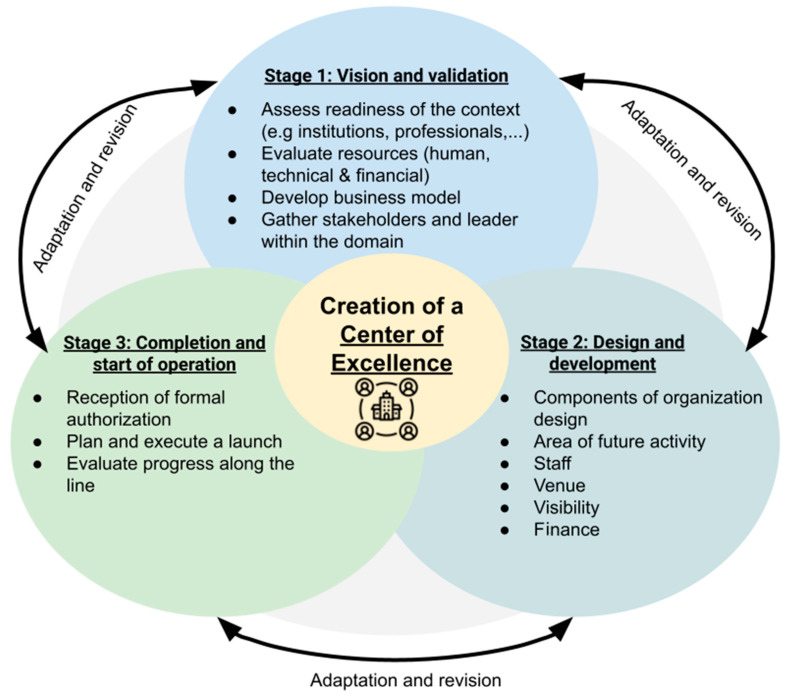
The 3-stage model of creation of a medication adherence research centre of excellence.

**Table 1 pharmaceutics-15-00933-t001:** The objectives of medication adherence research centres of excellence across various areas of their activities.

Area of Activity	Objectives
Research	To perform high-quality medication adherence-related research; To promote good practices in adherence research; To disseminate recent research findings in the area of medication adherence research at the local and international forum.
Clinical practice	To stimulate the adoption of evidence-based interventions toward adherence management; To introduce innovative approaches (e.g., eHealth) toward adherence management; To design clinical guidelines addressing medication adherence management.
Education	To provide under- and postgraduate education for healthcare professionals in medication adherence management; To stimulate nationwide multidisciplinary networking of various stakeholders active in the area of medication adherence; To provide patient education regarding medication adherence.
Advocacy & Policy	To raise awareness of non-adherence and promote available solutions among healthcare professionals and lay public; To identify and stimulate the adoption of the best evidence-based interventions and implementation strategies toward national policies; To support the adoption of effective and cost-effective solutions in daily clinical practice; To stimulate reimbursement of medication adherence-enhancing interventions; To promote the inclusion of medication adherence in educational curricula.

## Data Availability

Not applicable.

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
