# Peer review of "Half a Century of Fragmented Research on Deviations from Advised Therapies: Is This a Good Time to Call for Multidisciplinary Medication Adherence Research Centres of Excellence?"

_pharmaceutics, 2023, doi:10.3390/pharmaceutics15030933_

Round 1

Reviewer 1 Report

In this research, the authors proposed the structure of CoE (center for excellence) for assessment and control of medical adherence. 

It seems that the manuscript includes an appropriate architecture for the CoE, and the three compartment looks reasonable. I have one major concern about this research. In this research, the authors claimed that their experience supports their design of CoE; however, the experience was presented only in a descriptive manner, and the objective indices that supports the utility of the design was not introduced and reported. The results might be in the reference, but I think it is necessary to present the information in the manuscript to prove that the design is valid. 

Author Response

Dear Reviewer,

Thank you very much for this comment, which helped us to make the manuscript even better. In order to address this comment well, we have added an additional paragraph, under the title ‘Lessons learned’, which describes the way that two CoEs of interest have been designed, and the consequences of this process, as well as lessons learned for future. 

Reviewer 2 Report

The perspective of the authors is very interesting. The interest in researching medication adherence has been increasing in the last years and even if practical interventions were proposed, their implementation is delayed. The idea of the Medication adherence research Centers of Excellence was implemented in two countries and the outcomes are very well described.

I do not understand what represents the red color from Figure 1:  are there sums of all the articles with yellow and green? 

Why do you think these CoE will have a better impact on the health stakeholders? What were the difficulties the CoEs encountered and how were they handled?

How can be paid medication adherence interventions? There are a small number of paid medication adherence interventions, as you have already discussed in your manuscript.

Author Response

Dear Reviewer,

We would like to express our thanks for your comments. We hope that responding them, we have made our manuscript even better. Below, please find detailed aswerss to all your points.

  1. I do not understand what represents the red color from Figure 1: are there sums of all the articles with yellow and green?

Response: Yes, the red area represents the sum of green (corresponding with publications retrieved with the keyword ‘medication adherence’) and yellow (corresponding with publications retrieved with the keyword ‘patient compliance’). In order to make it more straightforward, the figure’s legend has been slightly modified.

  1. Why do you think these CoE will have a better impact on the health stakeholders?

Response: A response to this question has been provided in the paragraph which starts with “Along with these principal aims…”. In order to make this section more illustrative, it has been extended a bit.

  1. What were the difficulties the CoEs encountered and how were they handled?

Response: Thank you very much for this comment, which helped us to make the manuscript more complete. In order to address this comment well, we have added an additional paragraph, under the title ‘Lessons learned’, which – among the others – describes the challenges that two CoE met, and the solutions adopted.

  1. How can be paid medication adherence interventions? There are a small number of paid medication adherence interventions, as you have already discussed in your manuscript.

Response: Medication adherence interventions could be paid to the healthcare providers according to the various indicators. For details, please visit references No 13 & 14. However, we do not discuss this question in this very manuscript, being convinced that this question goes beyond the scope of it.

Round 2

Reviewer 1 Report

1. Please submit a manuscript after completion of proofreading. The current version is hard to read. 

2. In Fig. 2, 'FInance' seems to be changed to 'Fiance'.

3. (Optional) I'd like to suggest that the authors provide any measure to evaluate questions they proposed in the 'Lessons Learned' section. I think the tentative measures would help understanding the ideas of this manuscript. 

Author Response

Thank you for your useful comments! Here below, we address all the points raised by you:

Please submit a manuscript after completion of proofreading. The current version is hard to read. 

Answer: Clean version has been submitted.

In Fig. 2, 'FInance' seems to be changed to 'Fiance'.

Answer: Figure has been corrected.

(Optional) I'd like to suggest that the authors provide any measure to evaluate questions they proposed in the 'Lessons Learned' section. I think the tentative measures would help understanding the ideas of this manuscript. 

Recently, during the meeting of ENABLE group in Luxembourg, we had a discussion over the shape of CoEs. Therefore, we have use this discussion as a basis over which the 'lessons learned' have been selected. In order to better reflect above-mentioned discussion, the sequence of the relevant points have been changed, respectively.
